# Antibody–Drug Conjugates for the Treatment of Renal Cancer: A Scoping Review on Current Evidence and Clinical Perspectives

**DOI:** 10.3390/jpm13091339

**Published:** 2023-08-30

**Authors:** Stefano Sganga, Silvia Riondino, Giovanni Maria Iannantuono, Roberto Rosenfeld, Mario Roselli, Francesco Torino

**Affiliations:** Medical Oncology Unit, Department of Systems Medicine, University of Rome Tor Vergata, Via Montpellier 1, 00133 Rome, Italy; stefano.sganga@gmail.com (S.S.); silvia.riondino@uniroma2.it (S.R.); gmiannantuono@gmail.com (G.M.I.); roberto.rosenfeld88@gmail.com (R.R.); mario.roselli@uniroma2.it (M.R.)

**Keywords:** antibody–drug conjugates, ADC, renal cell carcinoma, monoclonal antibody, linker, cytotoxic payload

## Abstract

Antibody–drug conjugates (ADCs) are complex chemical structures composed of a monoclonal antibody, serving as a link to target cells, which is conjugated with a potent cytotoxic drug (i.e., payload) through a chemical linker. Inspired by Paul Ehrlich’s concept of the ideal anticancer drug as a “magic bullet”, ADCs are also highly specific anticancer agents, as they have been demonstrated to recognize, bind, and neutralize cancer cells, limiting injuries to normal cells. ADCs are among the newest pharmacologic breakthroughs in treating solid and hematologic malignancies. Indeed, in recent years, various ADCs have been approved by the Food and Drug Administration and European Medicines Agency for the treatment of several cancers, resulting in a “practice-changing” approach. However, despite these successes, no ADC is approved for treating patients affected by renal cell carcinoma (RCC). In the present paper, we thoroughly reviewed the current literature and summarized preclinical studies and clinical trials that evaluated the activity and toxicity profile of ADCs in RCC patients. Moreover, we scrutinized the potential causes that, until now, hampered the therapeutical success of ADCs in those patients. Finally, we discussed novel strategies that would improve the development of ADCs and their efficacy in treating RCC patients.

## 1. Introduction

### 1.1. Rationale

Renal cell carcinoma (RCC) accounts for 3–5% of all malignancies, being the sixth most frequently diagnosed cancer in women and the tenth in men [1]. Cytotoxic anticancer agents have been historically ineffective in RCC [2]. Moreover, initial immunotherapies, i.e., high-dose interleukin-2 (IL2) or α-interferonα (IFN), showed limited benefit in terms of overall response and survival [3,4]. Conversely, in the last decades, the treatment landscape of metastatic RCC (mRCC) has been revolutionized by highly effective agents, namely tyrosine kinase inhibitors (TKIs, i.e., sunitinib, pazopanib, cabozantinib, axitinib, lenvatinib, and tivozanib) and immune checkpoint inhibitors (ICIs, i.e., nivolumab, ipilimumab, and pembrolizumab), initially used as single agents and, more recently, as combination therapy [5]. These agents could improve tumor control and survival in randomized clinical trials and the real world [6]. According to Demasure et al., a significant improvement in overall survival in a real-world cohort of patients affected by mRCC was observed over the last 15 years, coinciding with the introduction of VEGFR-TKIs and ICIs. In detail, the median overall survival of patients was 13 months when the first-line treatment was IFN, 19 months when patients started with TKIs, and 45 months when the first-line treatment was based on ICIs (*p* < 0.0001). Five-year OS was 7%, 21%, and 36% in patients treated in the first line with IFN, TKIs, and ICIs, respectively [7].

Similarly, in a propensity-matched cohort study of 5872 patients treated in real-world clinical practice, first-line immunotherapy and combination therapies were associated with improved overall survival compared with first-line targeted therapy. The 12-month overall survival was 59% in the targeted therapy group, 73% in the immunotherapy group, and 68% in the combination targeted therapy and immunotherapy group [8]. However, despite the recent remarkable survival improvement achieved by new drugs and combinations, mRCC remains highly lethal. Approximately 134,000 deaths occur, compared with 295,000 new diagnoses per year, so it is critical to search for new anticancer agents for patients affected by mRCC [9,10,11].

In recent years, a novel class of anticancer drugs characterized by a “smart chemotherapy delivery” has been developed for the treatment of hematological and solid malignancies: the antibody–drug conjugates (ADC) [12,13,14,15]. These drugs are composed of a monoclonal antibody (mAb) bound to a cytotoxic drug (payload) through a linker. In this way, the drug selectively binds the target(s), i.e., antigens expressed on cancer cell surface, and is then internalized with its cytotoxic payload into the cell [16]. The idea behind ADCs is to produce drugs that selectively bind only neoplastic cells in order to be successively internalized into the tumor cell and then degraded in lysosomes [17]. Subsequently, ADC degradation will allow the release of the active payloads, which will be responsible for the induction of apoptosis, antibody-dependent cellular cytotoxicity (ADCC), and/or complement-dependent cytotoxicity (CDCC) (Figure 1) [17].

As a result, this class of drug will be able to stimulate apoptosis or necrosis of cancer cells, essentially by causing direct damage to the cell’s DNA or by inhibiting the formation or activity of intracellular microtubules [18]. In addition, the diffusion of the payload within the tumor microenvironment will also promote cellular damage in the adjacent neoplastic cells, the so-called “bystander effect” (Figure 2). Finally, by binding to the antigen, the Fab portion could stimulate the immune system cells to recognize cancer cells, activating an immune-mediated response against the same cancer cells [19,20].

ADCs have shown a progressive and steady improvement in terms of therapeutic efficacy and safety, leading to the expansion of the therapeutic armamentarium of several types of cancers, as witnessed by the approval by the Food and Drug Administration (FDA) and the European Medicines Agency (EMA) of five different ADCs for the treatment of patients with solid tumors; however, these do not include RCC (Table 1).

### 1.2. Objectives

Considering the rapid evolution of ADCs and their remarkable activity against multiple cancers resulting in approvals in different indications, we evaluated the available evidence on ADC as a potential treatment for patients affected by RCC. To this aim, a scoping review was conducted to scrutinize the available literature.

## 2. Materials and Methods

### 2.1. Protocol and Registration

The available literature was systematically reviewed according to the Preferred Reporting Items for Systematic reviews and Meta-Analyses (PRISMA-ScR) criteria for scoping reviews [31]. The protocol was designed a priori and, after being approved by all the authors, it was registered on the open science framework website (available at “https://osf.io/y8hkf/”) (accessed on 1 March 2023).

### 2.2. Eligibility Criteria

The PubMed database, from its inception to 5 June 2023, was searched. All publications that focused on preclinical or clinical studies on ADC in RCC patients were included. Only papers written in English were considered. The search strategy was established through a discussion among all the authors. The following syntax was used: “(antibody-drug conjugate OR antibody drug conjugate) AND (renal cancer OR kidney cancer OR renal cell carcinoma* OR renal cell cancer)”. The data obtained were uploaded to a reference management software (Zotero).

### 2.3. Selection of Sources of Evidence

A two-stage study selection process was used for the literature search, and it was performed independently by two of the authors. Firstly, titles and abstracts were initially screened for potential relevance. Secondly, full texts of relevant results were retrieved and further assessed for eligibility. A third author was required to resolve disagreements on study selection by consensus at both stages. Lastly, the two authors agreed on the publications to be included in the scoping review before beginning the data charting process.

### 2.4. Data Items and Synthesis of Results

Preclinical, phase I, and phase II studies were analyzed in the first section. In preclinical articles, the therapeutic target, the type of molecule used, the drug activity reported, and the publication date were evaluated. In phase I studies, the type of ADC used, its mechanism of action in vivo, the therapeutic target and expression levels in patients affected by RCC, and safety were assessed. In addition, in phase 2 studies, we examined the activity and results. Finally, in every article, primary and secondary outcomes, efficacy, adverse effects, tolerability for patients, and year of publication were considered.

## 3. Results

### 3.1. Selection of Sources of Evidence

The identification process of eligible publications and the results of the literature search were displayed using a flow diagram structured according to PRISMA guidelines [31]. Three hundred and sixty-four articles published between 1987 and 2023 were evaluated in this scoping review, but only twelve articles were considered eligible for our research. The results of single sources of evidence eligible for the present scoping review are described in Figure 3.

### 3.2. Synthesis of Results

Twelve articles were included in the systematic review. In particular, five preclinical studies, six phase I studies, and one phase II study were included. The main characteristics of the articles are summarized in Table 2.

### 3.3. Preclinical Studies

In the present review, we identified five preclinical studies in which different ADCs and different targets were tested.

#### 3.3.1. L49 and L6

The study of Svensson, published in 1998, evaluates the efficacy of two monoclonal antibodies (mAbs), L6 IgG_2a_ and L49 IgG_1_. These mAbs were chemically conjugated to Enterobacter cloacae-lactamase to activate cephalosporin containing anticancer prodrugs [32]. P97 was chosen as target of action of these mAbs, a protein hyper expressed in melanoma and several types of solid tumors, including RCC [44]. P97 is an iron (Fe) binding transferrin (Tf). Its role is still unclear; however, it seems to be involved in physiological and pathological processes, such as Fe transport Alzheimer’s disease and eosinophil differentiation [45,46]. In mouse models based on the use of tumor xenografts (mice with established SN12P or 1934J), L49 mAb demonstrated a higher affinity for the target and discrete treatment efficacy, while L6 mAb showed no significant benefit in terms of activity [32].

#### 3.3.2. HKT288

HKT-288, also called maytansine-based ADC HKT-288, is an ADC composed of a fully human IgG, a linker consisting of N-succinimidyl 4-(2-pyridyldithio)-2-sulfobutanoate (sulfo-SPDB), N2′-deacetyl-N2′-(4-mercapto-4-methyl-1-oxopentyl)-maytansine (DM4), and a maytansine-derived payload [33]. The target of this ADC is cadherin-6 (CDH6), a synaptic adhesion molecule located on the basolateral membrane of epithelial cells that interacts with calcium-dependent cell–cell adhesion. This molecule is highly expressed in serous ovarian carcinoma and RCC [47,48,49,50].

In 2017, HKT288 was evaluated in a preclinical study, in three different renal tumor models (HKIX3629, HKIX3717, and HKIX5374), and transplanted into nude mice. Administration of this ADC was associated with a significant antitumor effect in the HKIX3629 model (the one with higher CDH6 expression). These data encouraged the design of a phase I trial (NCT02947152), which was discontinued after the enrollment of nine patients (five with RCC and four with epithelial ovarian cancer), unfortunately demonstrating no significant clinical benefit [51].

#### 3.3.3. 138H11

138H11 is an ADC composed of a 138H11 mAb IgG1-DNA-cleaving enediyne calicheamicin thetaI1 (Camtheta) conjugate using SPDP (succinimidyl 3-(2-pyridyldithio) propionate) as a covalent linker [34]. The target of this mAb is represented by human renal anti gamma-glutamyltransferase (gamma GT). This molecule is widely expressed in clear cell and papillary RCC [52]. Even though this mAb has been evaluated both in vitro and in vivo, showing significant inhibition of tumor growth and moderate tolerability, and administration of 138H11-Camtheta was related to significant tumor growth inhibition, no further studies on the molecule were started.

#### 3.3.4. 1F6-vcAFP and 1F6-vcMMAF

1F6-vcAFP and 1F6-vcMMAF are two different ADCs with the same mAb (IgG1) but different payloads. The first one is conjugated with auristatin phenylalanine phenylenediamine (AFP), while the second is conjugated with monomethyl auristatin phenylalanine (MMAF) [35].

These ADCs are directed against CD70, a type II integral membrane protein generally expressed on T and B lymphocytes, natural killer NK, and mature dendritic cells [53]. High expression of CD70 is observed in clear cell and papillary RCC cells [54]. In vitro, both 1F6-vcAFP and 1F6-vcMMAF demonstrated a dose-dependent cytotoxic activity, caused by the destruction of the microtubule network and cell cycle arrest in the G2-M phase [35]. Moreover, in xenograft mouse models of Caki-1 and 786-O lines, the injection of 1F6-vcMMAF was correlated with a significant reduction in tumor mass in the absence of a particular adverse event [35]. However, no further studies on these two specific ADCs were found in the literature, and no clinical trials are ongoing.

#### 3.3.5. h1F6

The h1F6-mcMMAF is a humanized mAb conjugated to monomethyl auristatin F (MMAF), an anti-tubulin agent, via an uncleavable maleimidocaproyl linker, targeting the CD70 protein. In a study published in 2008 [36], MMAF was evaluated both in vitro, and in vivo and, in particular, in an experimental metastasis model UMRC-3, implanted i.p. in mice [37]. The researchers demonstrated that increasing doses of h1F6 correlated with a significant reduction in neoplastic mass. The results obtained in this study were the basis for a further phase I study (see below) [37].

### 3.4. Phase I Clinical Trials

According to our research results, ADCs were evaluated as an anticancer treatment in patients affected by RCC in only six phase I clinical trials.

#### 3.4.1. SGN-75 Anti-CD70

SGN-75 is a humanized anti-CD70 IgG1 monoclonal antibody, h1F6, also known as SGN-70, chemically conjugated to monomethyl auristatin F (MMAF), a synthetic analog of the natural microtubule-destroying agent, dolastatin 10, via a maleimidocaproyl (mc) [36,49]. This ADC was tested in a multicenter phase I dose-escalation study in patients with CD70-positive relapsed/refractory non-Hodgkin lymphoma and mRCC (NCT01015911) [37]. The role of CD70-positive cells in RCC patients is not known. However, a high expression of CD70 is believed to be related to mechanisms of immunotolerance of tumor cells against the host immune system. In fact, it appears that CD27-CD70 interaction increased the frequency of regulatory T cells (Tregs), reduced tumor-specific T cell responses, increased angiogenesis, and promoted tumor growth [55]. In this study, the primary outcome was to identify the safety and tolerability of SGN-75 and to evaluate the maximum tolerated dose (MTD) of SGN-75. Thirty-nine patients with RCC were enrolled and treated every 3 weeks (n = 32) or weekly (n = 7) with SGN-75 at different concentrations; overall, treatment was well tolerated, as the main complaints were G1-G2 adverse events. The best clinical responses of RCC patients treated every 3 weeks (Q3Wk) (n = 32) were the following: two patients treated with 2.0 and 3.0 mg/kg had a partial response (PR), and 12 patients treated with ≥1.5 mg/kg, had stable disease (SD). Conversely, for RCC patients treated Q3Wk who achieved SD or better (n = 14), the median duration of disease control was 46.4 weeks. The median PFS for RCC patients Q3Wk was 7.3 weeks. On the contrary, two SD and five PD were reported in patients treated weekly. The study demonstrated that an anti-CD70 ADC provides RCC patients with a discrete clinical benefit associated with poor and well-controlled adverse effects. This study has also suggested that a higher expression of CD70 is associated with a greater response to ADC.

#### 3.4.2. CDX-014 Anti-Immunoglobulin Mucin-1 TIM-1

CDX-14 is an ADC consisting of a human G1 (IgG1), covalently linked with the antimitotic monomethyl auristatin E (MMAE) and directed against Immunoglobulin Mucin-1 (TIM-1). This ADC was evaluated in a multicenter, phase I, open-label, single-arm study evaluating the safety and preliminary activity in patients affected by advanced refractory RCC or ovarian carcinoma (NCT02837991) [38]. The target of this ADC is T cell Immunoglobulin Mucin-1 (TIM-1), also known as Kidney Injury Molecule 1 (KIM-1), a transmembrane glycoprotein largely expressed at the level of renal tubule cells because of ischemic processes and/or renal damage. A recent study has shown that TIM-1 is expressed by the cells of all major renal tumors [56].

In this study, 16 patients with advanced clear cell or papillary RCC were enrolled, who received a median of four cycles of CDX-014. The study showed that 31% of patients exhibited clinical benefits from CDX-014, with PFS and OS of 2.7 months (95% CI, 1.2–8.0) and 12.6 months (95% CI, 5.7–12.6), respectively. In addition to the discrete efficacy, this study has demonstrated the good tolerability of the treatment. In fact, during the study, only two events of dose-limiting toxicities (i.e., liver dysfunction and hyperglycemia) were registered. Despite those encouraging results, no further studies have started on CDX-14.

#### 3.4.3. SGN-CD70A Anti-CD70

SGN-CD70A consists of a mAb (IgG1) conjugate DNA-crosslinking pyrrolobenzodiazepine (PBD) dimer drug, via a cleavable linker, and directed against CD-70 (NCT02216890) [39]. PBD was tested in several trials with encouraging results [57]. In this study by Pal and coworkers, 18 mRCC patients and 20 non-Hodgkin lymphoma (NHL) patients were enrolled. The primary outcomes were the detection of the incidence of adverse events and the incidence of laboratory abnormalities to identify the MTD of SGN-CD70A. All RCC patients were required to have at least 50% CD70A expression (defined as CD70-positive RCC). The treatment proved to have a modest efficacy associated with manageable adverse effects. In particular, PR was detected in only one patient. On the other hand, in 72% of patients, SD was observed as the best clinical response.

#### 3.4.4. MDX-1203 Anti-CD70

BMS-936561, also called MDX-1203, is composed of human IgG1 conjugated with a MED-2460 or duocarmycin molecule through a peptide-based linker directed against CD70, and it was evaluated in patients with advanced clear cell RCC (ccRCC) and B lymphocyte-NHL (B-NHL) (NCT00944905) [40].

In this study, 26 patients with ccRCC or B-HNL were enrolled to determine the safety profile and MTD as primary outcomes. The treatment was well tolerated, with few adverse effects. However, such adequate tolerability and manageability were associated with moderate treatment efficacy. In fact, 18 of the 26 (69%) subjects achieved SD, which was more frequent in patients receiving a higher dose. In contrast, no patient achieved PR or a complete response (CR).

#### 3.4.5. AMG 172 Anti-CD70

AMG 172 is an all-human IgG1, anti-CD 70, with a DM1-based payload (a semisynthetic derivative of the antibiotic ansamycin, maytansine), conjugated to mAb via a non-cleavable linker 4(-[N-maleimidomethyl]cyclohexane-1-carboxylate) (NCT01497821) [41]. The study by Massard and coworkers enrolled thirty-seven patients exclusively with ccRCC. The primary outcome was to assess safety, pharmacokinetics parameters, MTD, and the objective response rate for subjects treated at MTD. The data obtained revealed two cases (5.4%) of PR. On the other hand, in six patients (16.2%), SD was obtained, but it only remained stable in one patient at subsequent radiological re-evaluations. Finally, 13 patients (35.1%) had PD. Unfortunately, it was not possible to evaluate 16 patients according to RECIST version 1.1 due to insufficient follow-up scans.

This study has demonstrated a fair clinical efficacy associated with overall good tolerability.

#### 3.4.6. AGS-16M8F and AGS-16C3F Anti-Phosphodiesterases-Pyrophosphatase 3 (ENPP3)

GS-16M8F and AGS-16C3F are two ADCs composed of fully human IgG2a conjugated to MMAF via a non-cleavable linker [42,58]. Their target is ectonucleotide pyrophosphatase/phosphodiesterase 3 (ENPP3: CD203a), a member of the ENPP family. ENPP3 is a marker of basophil activation, and it is composed of the extracellular domain of ENPP3, a type II transmembrane protein, a catalytic domain, and a somatomedin B-like domain [59]. However, the role of ENPP3 in RCC carcinogenesis is unknown. Nevertheless, the high ENPP3 expression in neoplastic cells makes it a conceivable therapeutic target [60,61]. In particular, it has been found in patients with ccRCC and, to a lesser extent, in patients with papillary RCC [58].

The first anti-ENPP3 mAb for the treatment of RCC was hybridoma-derived AGS-16M8F; in consideration of the high cost and long time to produce antibodies via hybridomas, a second antibody directed against ENNP3, based on the Chinese Hamster Ovary (CHO) cell line system, AGS-16C3F, was derived [58]. In 2018, the results of two sequential phase I studies were published based on the administration of these ADCs, (NCT01114230) and (NCT01672775) [42]. For both studies, the primary outcome was to evaluate the safety of AGS-16M8F(Hyb)/AGS-16C3F(CHO). All patients tolerated the treatment well, even if approximately half of the subjects reported ocular symptoms such as dry eye (17/34, 50%) and blurred vision (15/34, 44%). Interestingly, GS-16C3F(CHO) obtained a PR in 3 out of 13 patients that lasted between 100 and 143 weeks. Additional data were obtained from a phase II study (see below).

### 3.5. Phase II Clinical Trials

Currently, the only one phase II trial on ADC in RCC patients has resulted in termination. In that trial, the impact of the AGS-16C3F on PFS was compared with axitinib [43]. To the best of our knowledge, no further phase II clinical trials evaluating ADCs in patients with RCC are ongoing.

#### AGS-16C3F versus Axitinib

In this study, the safety and efficacy of AGS-16C3F were evaluated versus axitinib as primary outcomes in previously treated patients with mRCC of any histology (NCT02639182) [43]. In this study, 133 patients were enrolled and were randomized 1:1 to either intravenous AGS-16C3F 1.8 mg/kg every 3 weeks or oral axitinib 5 mg twice daily. This study revealed a PFS of 2.9 months in patients treated with AGS-16C3F, vs. 5.6 months with axitinib. Therefore, the primary endpoint of the study was not met, and no significant differences between AGS-16C3F and axitinib in secondary endpoints were detected. In particular, the median (OS) was 13.1 months with AGS-16C3F and 15.4 months with axitinib. However, both drug treatments have shown a good tolerability and safety profile.

## 4. Discussion

### 4.1. Summary of Evidence

The recent advances in genomics and molecular biology, along with the clarification of the cancer-immunity cycle, have profoundly changed the treatment paradigm of mRCC [62]. In the last two decades, the therapeutical management of patients affected by this malignancy has evolved from immune cytokine-based approaches (i.e., IL2 and IFN-α) to targeted therapies consisting of TKI against either vascular endothelial growth factor-receptor (VEGF-R), mammalian target of rapamycin (mTOR), or anaplastic lymphoma kinase (ALK) [63,64]. Moreover, the recent advent of immunotherapy with monoclonal antibodies modulating some immune checkpoints (ICI) has represented a novel breakthrough in the immunological treatment of RCC [62].

More recently, phase III clinical trials have demonstrated a significant survival benefit for patients with advanced RCC receiving an ICI–ICI or ICI–TKI combination as a first-line treatment, while the best therapeutic strategy for second and further line(s) of treatment remains unclear [65,66,67]. However, despite the increased availability of effective anticancer agents and the remarkable improvements in survival obtained by therapeutic approaches combining effective drugs, the resistance of RCC cells to current drugs leads patients to death. It appears to be, therefore, essential to evaluate novel anticancer agents that could further enhance patient clinical outcomes.

ADCs represent a novel category of anticancer agents designed to obtain a targeted delivery of cytotoxic agents to cancer cells [63]. They are constituted by three different elements: a cytotoxic drug (termed a ‘payload’) conjugated to a monoclonal antibody targeting an antigen of interest that is specifically expressed on cancer cells. Indeed, ADCs exert anticancer activity by delivering the payload to cancer cells through the formation of an antibody–antigen complex, followed by receptor-mediated endocytosis. ADCs, merging the specificity of monoclonal antibodies with the potency of highly cytotoxic agents, potentially reduce the severity of side effects by preferentially targeting their payload to the tumor site [68]. However, ADCs are variably charged by off-target toxicities resembling those triggered by the cytotoxic payload, including fatigue, alopecia, cytopenias, gastrointestinal disturbances, and on-target toxicities [68,69]. Notably, unconventional (e.g., ocular toxicity) and potentially life-threatening toxicities can also be observed with certain ADCs, requiring a better understanding of these adverse events’ mechanisms [68,69,70]. A broader knowledge of ADC toxicity profiles among oncologists is essential for improving early diagnosis of toxicities and the related management practices [69,70]. Moreover, the availability of predictive factors of ADC-related toxicities will improve both the clinical use and the development of ADCs.

Nowadays, several ADCs are approved for the treatment of patients affected by hematological and solid malignancies. Currently, the use of ADCs is progressively expanding the therapeutical armamentarium of patients affected by HER-2 positive breast cancer. First, the use of Trastuzumab emtansine (T-DM1) and, then, of Deruxtecan trastuzumab have obtained significant improvements of OS and PFS in patients affected by metastatic breast cancer [22,26]. Similarly, in triple-negative breast cancer, the use of ADCs is also demonstrating clear benefits. Specifically, the use of sacituzumab govitecan (SG) improved both OS and PFS, in metastatic and multi-drug treated patients, allowing its approval in patients who have received two or more prior systemic therapies by FDA and EMA [28]. Moreover, in 2021, the FDA also approved SG for the treatment of patients affected by locally advanced or metastatic urothelial cancer who previously received a platinum-containing chemotherapy [29]. SG has also been tested in the treatment of metastatic non-small-cell lung cancer, achieving interesting results in the absence of relevant adverse effects [71]. Notably, in bladder cancer, the discovery of new targets paved the way to various clinical trials that explored the use of ADCs directed against Nectin-4, Trop-2, or HER-2 with significant clinical benefits in terms of overall response and survival [72,73].

In this scenario, we aimed to critically revise the available literature to evaluate the potential role of ADC as a treatment for patients affected by RCC. Our review highlights various negative findings. Indeed, not even one ADC has yet been approved for the treatment of patients affected by RCC. More importantly, to the best of our knowledge, no ADCs are being evaluated in a phase II or III trial [74]. In the only available phase II study [43], the activity of AGS-16C3F, an ADC directed against ENPP3, was compared to an axitinib-based treatment, failing to achieve the primary outcome (AGS-16C3F was supposed to obtain a better PFS over the standard treatment). Similarly, other phase I trials that evaluated various ADCs in RCC patients did not show results that led to the design of further phase II trials.

Currently, only a phase I study (NCT05293496) is evaluating vobramitamab duocarmazine (MCG018), an ADC directed against B7-H3 (also referred to as CD276), a member of the B7 family of immune regulatory proteins, expressed in numerous solid tumors, including RCC. Interestingly, B7-H3 overexpression appeared to be correlated with an increased risk of recurrence, greater aggressiveness of the disease, and higher mortality in RCC. Unfortunately, the study has not yet generated searchable data [75].

The disappointing paucity of studies and the reported negative findings in patients affected by RCC may be related to some aspects. It is well known that the activity/efficacy of ADCs is proportional to the expression levels of their targets on the surface of cancer cells, the affinity between the mAb and the target, the payload’s activity, and the bystander effect’s grade of activation [76].

Presumably, ADC activity failure seen in RCC can be, firstly, attributed to the absence of a “robust” target on cell surface of RCC. Up to now, mAbs targeting CD70, CDH6, and ENPP3 have been developed and evaluated in phase I studies on patients affected by various malignancies, including RCC. The poor results obtained clearly suggest that the current ADC targets expressed on renal carcinoma cells might be either heterogeneous, or weak, or both.

It is therefore essential to identify highly sensitive, specific, and largely expressed, mutated, or amplified, antigens on renal cancer cells. To this aim, new technologies based on RNA-sequencing and protein-expression, already exploited to identify new antigens in breast cancer, are expected to be applied in RCC [77].

Moreover, negative results achieved by ADC use in RCC patients could be explained by certain mechanisms of resistance to ADCs. Indeed, despite not being fully understood, the main mechanisms of resistance to ADCs are supposed to be related to (a) the reduction of antigen expression level or the change of intracellular transport pathway, (b) the resistance to payloads, and (c) the production of antibodies against ADCs by the immune system (or combined factors) [73]. Mainly, the antigen expression levels, altered intracellular transport pathways, or innate resistance to payloads in renal cancer cells might have a role in hampering ADC activity in RCC patients. Remarkably, renal cancer cells possess a high quantity of molecules functioning as drug efflux pumps. It is well known that one of the most frequent mechanisms of resistance to ADCs is increased expression of drug efflux pumps [78]. This might render RCC intrinsically refractory to the action of ADCs (even in presence of “good” targets). Another possible explanation for the limited ADC activity in RCC might be related to the well-known refractory of RCC to cytotoxic anticancer agents. This implies that payloads different from cytotoxic might be more effective, such as targeted anticancer agents and immunotherapy, two highly effective therapeutical approaches in RCC patients in terms of both OS and ORR [10,11]. An indirect support to the latter explanation derives from a recent promising approach consisting of combinations of ADCs with other immunotherapies or targeted agents [79]. Interestingly, the addition of ADCs to ICIs may increase the recruitment of CD8þ effector T cells to tumor tissues, improving clinical response. Various clinical studies evaluating combinations of T-DM1 or brentuximab vedotin with inhibitors of Programmed Death-1 (PD-1) or its ligand PD-L1 (Clinical trials identifiers: NCT02924883, NCT03032107 and NCT01896999) are ongoing. In this light, the development of biparatopic antibodies has begun in recent years, which have the peculiarity of being able to bind two different epitopes or antigens, enabling them to simultaneously block two distinct targets or mediate effector functions, including activation of cytotoxic T and NK cells to induce tumor lysis [80]. In addition, site-specific ADCs are being developed that are characterized by conjugating small functional molecules to specific sites in antibody molecules, including cysteine, glutamine, unnatural amino acids, short peptide tags, and glycans. The obtained antibody–drug conjugates showed high homogeneity and potent antitumor activity both in vitro and in vivo [81].

Finally, ADCs are structurally engineered to activate the immune system. In particular, the Fc portion of mAbs triggers antibody-dependent cell-mediated cytotoxicity (ADCC) and antibody-dependent cell-mediated phagocytosis (ADCP). As an example, an innovative class of drugs, termed immune-stimulating antibody conjugates (ISACs), which are designed to bind neoplastic cells and activate antigen-presenting cells (APCs) in the tumor microenvironment, has recently shown promising results [82,83]. In addition, one of the mechanisms of resistance to immunotherapy by solid tumors is the formation of a dense cell stroma, which reduces the infiltration of immune cells into the tumor core. Therefore, the combination of immunotherapy with cytotoxic anticancer drugs or ADCs could promote greater antitumor activity, and several ongoing studies are evaluating the efficacy of treatments based on synergy immuno-oncology drugs with different ADCs [84]. One example is represented by the treatment of breast cancer, where the administration of immunotherapy alone has not been correlated with particular clinical benefits, while immunotherapy combined with an ADC in patients with PDL-1 positive cancer subtypes resulted in a higher PFS compared to ADC in monotherapy (phase II KATE2 T-DM1 plus either atezolizumab or placebo) [85]. The efficacy of the ICI/ADC combination is also being evaluated in urothelial cancers, where the combination of pembrolizumab with enfortumab vedotin has achieved remarkable results, allowing the FDA, in February 2020, to allow the drug combination as a breakthrough therapy in the first-line setting for patients with cisplatin-ineligible disease [86].

Although our search did not provide encouraging data on the therapeutic role of ADCs in RCC, the most recent evidence let us hypothesize that ADCs might be a valuable tool for the treatment of patients affected by RCC in the near future, once the abovementioned hurdles are resolved.

### 4.2. Summary of Evidence

The main limitation of this scoping review resides in the extraction of data from preclinical, phase I, and phase II studies published over more than 20 years. During this time frame, ADCs have presented numerous developments, both in the treatment of different malignancies and in the design of different linkers and payloads. Therefore, we decided to include only articles that were descriptions of available studies with a proper methodology and relevant data.

## 5. Conclusions

ADCs represent an innovative pharmacological approach that has widened the therapeutic options for patients with an increasing number of malignancies. Unfortunately, the same success has not been achieved for patients affected by RCC. Indeed, no ADC is currently approved for the treatment of these patients. Furthermore, no ADC is presently under evaluation in phase II–III clinical trials. The only exception remains vobramitamab duocarmazine (MCG018), an ADC directed against B7-H3 (CD276), which is being evaluated in a phase I trial enrolling several malignancies, including RCC. Despite the disappointing role of ADCs as an anticancer treatment in RCC patients that emerged from our research, we believe that some modified approaches could achieve better results even in these patients. Firstly, this could be achieved by selecting sensitive targets on renal cancer cells and designing highly specific mAbs against them. For this purpose, CD70, CDH6, and ENPP3 are suggested as promising markers. Secondly, assessing the high expression of these targets in selected patients could represent a valid strategy to improve the efficacy of ADCs. Finally, considering the recognized chemo-insensitivity of RCC and the limits of current treatments, including combinations of TKIs and immunotherapy, mRCC remains a disease with an unfavorable prognosis. Therefore, developing new therapeutic approaches, such as those offered by ADCs, by exploiting novel payloads and non-classical antibody formats, along with combinations with effective treatments, represents an intriguing field of research that has obtained promising results in other solid tumors [85,87].

## Figures and Tables

**Figure 1 jpm-13-01339-f001:**
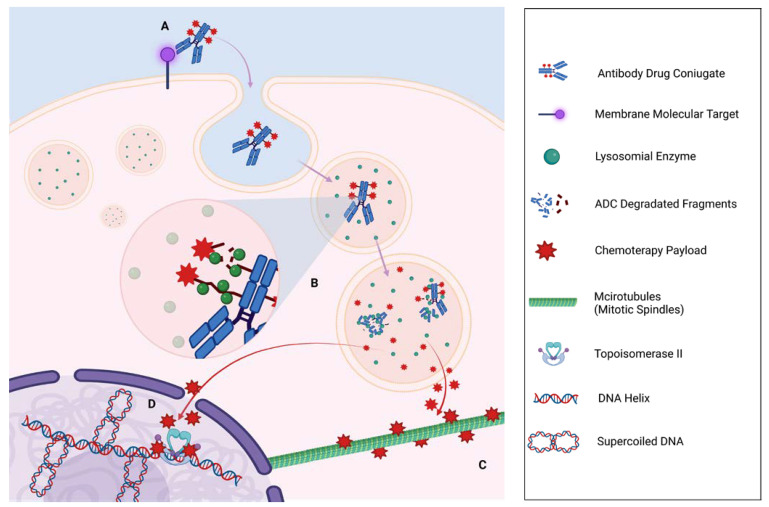
ADC Antibody Drug Conjugated (ADC) mechanisms of action in a cell cancer. (**A**) The Antigen-ADC binding induces phagocytosis of the monoclonal antibodies (mAB) through a Clatryn-coated endosome that subsequently fuses with lysosomes, initiating a proteolytic degradation of mAB through endopeptidases activities. (**B**) With the activation of lysosomal enzymes, linker degradation is initiated for cleavable mABs, or the entire mAB degradation is activated for non-cleavable bonds. (**C**) The final result is the release of the payload in the cytoplasm, which reaches and inhibits microtubules and Spindle cells, striking actively reproducing cells (e.g., T-DM1). (**D**) On the other hand, another important mechanism is the interference with Topoisomerase II function, resulting in a progressive accumulation of supercoiled DNA that prevents DNA polymerase from performing proper DNA duplication.

**Figure 2 jpm-13-01339-f002:**
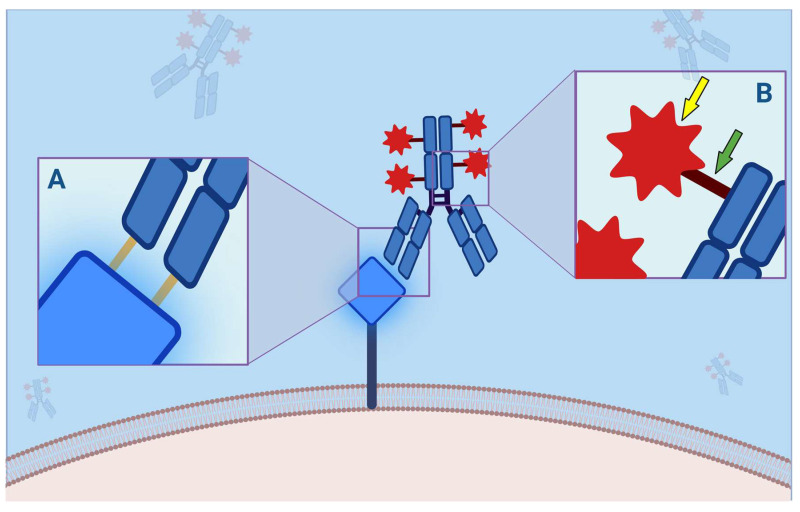
Antibody–Drug Conjugate (ADC) binding to its molecular membrane target (MMT). (**A**) Each ADC, as a monoclonal antibody, consists of a crystallizable fragment (Fc) and an antigen-binding fragment (FAB) portion, the latter consisting of 2 light and 2 heavy chains, each of which has a constant and a variable domain, which is responsible for the binding affinity to the mutated protein (orange bars). This interaction is critical because antigenic mutations account for resistance to ADCs. (**B**) The payload (yellow arrow) is bound to the Fc through a linker (green arrow). This binding can be divided into cleavable and not cleavable. Indeed, while the latter are more stable in the bloodstream, and their degradation is not possible without specific proteolytic enzymes, the cleavable ones are divided in acid-sensitive, protease-sensitive, or glutathione-sensitive and are more susceptible to hydrolysis. This mechanism is closely related to the “bystander effect”: the more easily a payload is released, the greater the clinical effect, such as adverse effects. On the other hand, non-cleavable ligands provide higher half-lives and fewer toxic effects.

**Figure 3 jpm-13-01339-f003:**
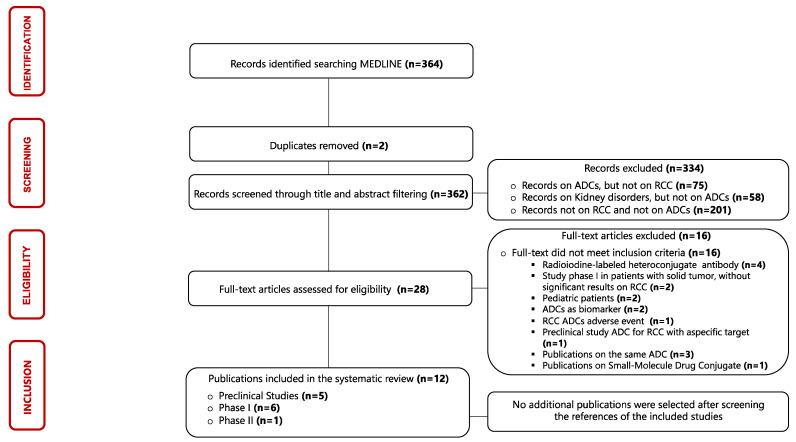
Flowchart of the results of the literature search and identification process of included publications. Abbreviations: ADC: antibody–drug conjugate; RCC: renal cell carcinoma.

**Table 1 jpm-13-01339-t001:** ADCs approved by FDA and EMA.

					Approval			
ADC	Trade Name (Pharma. Industry)	Target Antigen	Linkers	Payloads	FDA	EMA	Indications	**Clinical Trial (NCT)**
Ado-trastuzumab Emtansine [21,22,23]	Kadcyla (Roche)	HER-2	SMCC	DM1	X	X	Adjuvant treatment for HER-2 positive eBC patients with residual invasive disease after neoadjuvant taxane and trastuzumab-based treatment	KATHERINE Trial (NCT01772472)
					X	X	Locally advanced or metastatic HER-2 positive BC previously treated with taxane and trastuzumab	EMILIA Trial (NCT00829166)
					X	-	Unresectable or metastatic HER2-positive BC	DESTINY-Breast03 (NCT03529110)
Enfortumab vedotin [24]	Padcev (Seagen)	Nectin-4	mc-VC-PABC	MMAE	X	X	Locally advanced or mUC that have been previously treated with platinum chemotherapy and a PD-L1/PD-1 inhibitor	EV-301 Trial (NCT03474107)
Fam-trastuzumab Deruxtecan [25,26]	Enhertu (Daiichi Sankyo)	HER-2	Tetrapeptide	DXd	X	X	Unresectable or HER-2 positive mBC previously treated with two or more anti-HER2-based regimens	DESTINY-Breast01 (NCT03248492)
					X	-	Locally advanced or metastatic HER2-positive gastric or gastroesophageal junction adenocarcinoma who have received a prior trastuzumab-based regimen	DESTINY-Gastric01 (NCT03329690)
Sacituzumab govitecan [27,28]	Trodelvy (Immunomedics)	Trop-2	CL2A	SN38	X	X	Unresectable, locally advanced, or metastatic TNBC that have received two or more prior systemic therapies, at least one of them for metastatic disease	ASCENT (NCT02574455)
					X	-	Locally advanced or mUC that previously received a platinum-containing chemotherapy and either a PD-1 or a PD-L1 inhibitor	TROPHY-U-01 (NCT03547973)
Tisotumab vedotin [29]	Tivdak (Genmab/Seagen)	TF	mc-VC-PABC	MMAE	X	-	Recurrent or metastatic cervical cancer progressed on or after chemotherapy	innovaTV 204/GOG-3023/ENGOT-cx6 (NCT03438396)
Mirvetuximab soravtansine-gynx [30]	Elahere (ImmunoGen)	folate receptor alpha	sulfo-SPDB	DM4	X	-	Metastatic ovarian cancer that previously received platinum-based chemotherapy and have received 1 to 3 prior types of chemotherapy	MIRASOL trial (NCT04209855)

Abbreviations: eBC: early breast cancer; BC: breast cancer; mUC: metastatic urothelial carcinoma; mBC: metastatic breast cancer; TNBC: metastatic triple-negative breast cancer; X: approved by FDA or EMA; -: not approved by FDA or EMA.

**Table 2 jpm-13-01339-t002:** Preclinical and clinical trials on ADCs in patients affected by renal cell cancer.

ADC	Phase	mAb	Linker (Cleavable/Non-Cleavable)	Payload	Target	NCT
L49 L6 [32]	Preclinical	IgG2a IgG1	Valine-citrulline (CL)	CCM/CCM	p97	-
HKT288 [33]	Preclinical	IgG1	Sulfo-SPDB (CL)	DM4	CDH6	-
138H11 [34]	Preclinical	IgG1	SPDP (CL)	Camtheta	GGT	-
1F6 [35]	Preclinical	IgG1	Vc/pab (NCL)	vcAFP/vcMMAF	CD 70	-
h1F6 [36]	Preclinical	IgG1	MC (NCL)	MMAF	CD 70	-
SGN-75 [37]	Phase I	IgG1	MC (NCL)	MMAF	CD 70	NCT01015911
CDX-014 [38]	Phase I	IgG1	Valine-citrulline (CL)	MMAE	TIM-1	NCT02837991
SGN-CD70A [39]	Phase I	IgG1	Maleimidocaproic Valine-alanine (CL)	PBD	CD 70	NCT02216890
MDX-1203 [40]	Phase I	IgG1	Valine-citrulline (CL)	Duocarmycin	CD 70	NCT00944905
AMG 172 [41]	Phase I	IgG1	MCC (NCL)	DM1	CD 70	NCT01497821
AGS-16M8F AGS-16C3F [42]	Phase I	IgG2a/IgG2a	MC (NCL)/MC (NCL)	MMAF/MMAF	ENPP3	NCT01114230/ NCT01672775
AGS-16C3F [43]	Phase II	IgG2a	MC (NCL)	MMAF	ENPP3	NCT02639182

Abbreviations: ADC: antibody–drug conjugates; Camtheta: calicheamicin thetaI1; CCM: cephalosporin mustard; CD 70: cluster of differentiation; CDH6: cadherin-6; CL: cleavable linker; DM1: mertansine; DM4: ravtansine; ENPP3: ectonucleotide phosphodiesterases-pyrophosphatase 3; GGT: g-glutamyl transferase: mAb: monoclonal antibody; MC: maleimidocaproyl; MCC: 4-[N -maleimidomethyl] cyclohexane-1-carboxylate; MMAE: monomethyl auristatin E; MMAF: monomethyl auristatin F; NCL: non-cleavable linker; PBD: pyrrolobenzodiazepine; SPDP: succinimidyl 3-(2-pyridyldithio)propionate); sulfo-SPDB: N-succinimidyl 4-(2-pyridyldithio)-2-sulfobutanoate; TIM-1: Immunoglobulin Mucin-1; vcAFP: maleimidocaproyl valine-citrulline auristatin phenylalanine phenylenediamine; vcMMAF: maleimidocaproyl valine-citrulline monomethyl auristatin phenylalanine; vc/pab: valine-citrulline/p-aminobenzoic acid.

## Data Availability

Not applicable.

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
