# Peer review of "Antibody–Drug Conjugates for the Treatment of Renal Cancer: A Scoping Review on Current Evidence and Clinical Perspectives"

_jpm, 2023, doi:10.3390/jpm13091339_

Round 1

Reviewer 1 Report

1. Figure 1: The quality of figure need improvement. It is blurred.

2. Introduction: Some detail regarding current first line treatment of RCC and survival rate must be added.

3. Table 1: Heading Approval: FDA and EMA. what x indicates and what empty box indicates, it should mention.

4. Table 1: Column Indication, first Row: Pts should full form as Patients.

5. Result: Line 145 complete the sentence. "while L6mAb did not any effectiveness. 

6. Results: Figure 3: Why Figure 3 is part of result. Kindly explain as this kind of mechanistic details are already known.

6. Discussion: Line 322-325: There is mentioned that Phase III trails are conducted on RCC with ICI-ICI and ICI-TKI...[55, 56, 57]. 

Why these three references are not part of Figure 2 and Table 2. Table 2 shows 12 references showing Preclinical, and Phase I and Phase II Trails?

7. Conclusion: Why authors emphasisng future role of ICI and TKI, its not clear. Kindly elaborate.

Reviewer 2 Report

Thank you for the opportunity to review this manuscript focused on antibody-drug conjugates (ADCs) addressing the current challenges of Renal cell carcinoma (RCC).

I commend the authors for a well-compiled review of the clinical study results; it is genuinely well-written. However, it would enhance the comprehensiveness of this review if the general challenges associated with ADCs were also addressed, particularly common toxicities such as Neutropenia and ocular toxicity.

Furthermore, I am keen to hear about the future directions in this field. I anticipate discussions on advancements like the use of biparatopic antibodies, ISAC, Site-specific ADCs, and advancements in Linker-technology.
